# “To Be or Not to Be Benign” at Partial Nephrectomy for Presumed RCC Renal Masses: Single-Center Experience with 195 Consecutive Patients

**DOI:** 10.3390/diseases11010027

**Published:** 2023-02-07

**Authors:** Raffaele Baio, Giovanni Molisso, Christian Caruana, Umberto Di Mauro, Olivier Intilla, Umberto Pane, Costantino D’Angelo, Antonio Campitelli, Francesca Pentimalli, Roberto Sanseverino

**Affiliations:** 1Department of Medicine and Surgery “Scuola Medica Salernitana”, University of Salerno, 84081 Baronissi, Italy; 2Department of Urology, Umberto I, Nocera Inferiore, 84014 Salerno, Italy; 3Department of Chemistry, University of Malta, MSD2080 Msida, Malta; 4Department of Medical Biotechnologies, University of Siena, 53100 Siena, Italy; 5Department of Medicine and Surgery, LUM University, 70010 Bari, Italy

**Keywords:** laparoscopic partial nephrectomy, benign renal mass, tumor size

## Abstract

In daily medical practice, an increasing number of kidney masses are being incidentally detected using common imaging techniques, owing to the improved diagnostic accuracy and increasingly frequent use of these techniques. As a consequence, the rate of detection of smaller lesions is increasing considerably. According to certain studies, following surgical treatment, up to 27% of small enhancing renal masses are identified as benign tumors at the final pathological examination. This high rate of benign tumors challenges the appropriateness of surgery for all suspicious lesions, given the morbidity associated with such an intervention. The objective of the present study was, therefore, to determine the incidence of benign tumors at partial nephrectomy (PN) for a solitary renal mass. To meet this end, a total of 195 patients who each underwent one PN for a solitary renal lesion with the intent to cure RCC were included in the final retrospective analysis. A benign neoplasm was identified in 30 of these patients. The age of the patients ranged from 29.9–79 years (average: 60.9 years). The tumor size range was 1.5–7 cm (average: 3 cm). All the operations were successful using the laparoscopic approach. The pathological results were renal oncocytoma in 26 cases, angiomyolipomas in two cases, and cysts in the remaining two cases. In conclusion, we have shown in our present series the incidence rate of benign tumors in patients who have been subjected to laparoscopic PN due to a suspected solitary renal mass. Based on these results, we advise that the patient should be counseled not only about the intra- and post-operative risks of nephron-sparing surgery but also about its dual therapeutic and diagnostic role. Therefore, the patients should be informed of the considerably high probability of a benign histological result.

## 1. Introduction

The use of common and noninvasive imaging techniques (including the use of computed tomography (CT), ultrasonography, and magnetic resonance imaging (MRI)), which has been increasing exponentially over the last few decades, has led to a similarly increased detection rate of small, localized, and often asymptomatic kidney tumors [1], involving a risk of cancer overdiagnosis [2]. Furthermore, a number of these tumors are shown to be benign following definitive pathological examination, and, to date, there are no specific imaging findings that may be used to uniquely diagnose a tumor as malignant or benign [3,4]. According to Remzi et al. [5], only 17% of renal masses are correctly classified as benign on performing preoperative CT. A total of 43% of these masses were subsequently overtreated with radical nephrectomy. Benign tumors are not uncommon, even in the case of renal masses with a diameter > 4 cm, and, if large, surgical treatment of these benign tumors is also associated with considerable morbidity [6]. This scenario has enabled a choice of partial nephrectomy (PN) for the surgical treatment of these masses. As a result, starting from 2009, both American and European guidelines have recommended PN as the gold standard for patients with T1 masses that can be excised in a feasible manner [7] and, from 2014, this indication has also evolved to include T2 lesions that are technically amenable to PN. In effect, compared with radical treatment, elective PN offers similar oncological outcomes, especially for those patients who are diagnosed with stage T1 renal cell carcinoma (RCC) [8,9,10,11]. In their study that included and assessed patients treated with PN (379) or radical nephrectomy (1075) for T1N0M0 kidney tumors, Patard et al. [11] demonstrated that no significant differences existed in the local or distant recurrence rates in halfway medical checks of >5 years. In their study, approx. 85% of the PNs were performed for renal masses ≥ 4 cm. According to several studies, 20–30% of small kidney tumors are benign at pathological examination after surgery, despite expert preoperative radiological estimations [3,9,10,12]. For this reason, considering the morbidity associated with nephron-sparing techniques (regardless of whether they are open, laparoscopic, or robot-assisted), the appropriateness of this type of surgery for all masses suspected to be renal tumors must be challenged. In effect, according to the results of a review of nine studies that included >1000 patients, performing PN led to low but fundamentally operative morbidity and mortality rates [4]. The types of complications, and their incidence rate, based on this review [4], are: urinary fistula (7.4%), infection or abscess (3.2%), postoperative bleeding (2.8%), reintervention (1.9%), and perioperative death (1.6%). Another study assessing 180 patients who underwent PN [13,14,15] reported, following surgery, hemorrhage in four (2.2%) patients and urinary fistula in an additional three (1.7%) patients. Consequently, even though PN can be considered a valid method for both the diagnosis and treatment of small kidney lesions, its suitability for all suspicious masses remains questionable; moreover, its widespread use has been debated, given the great impact of benign tumors and the morbidity rate that is associated with this surgical procedure. Therefore, given current guidelines that recommend treatment unless the patient is elderly or infirm, the overtreatment of patients with benign renal tumors remains an inappropriate risk. For all the above reasons, the purpose of the present study was to investigate the incidence of benign tumors at the stage of receiving PN on preoperative imaging evaluations in a single-center series of patients with a solitary renal lesion considered to be RCC.

## 2. Patients and Methods

The medical records of patients receiving PN at our center (Department of Urology, Umberto I Hospital) over the last 10 years were retrospectively reviewed. Elective PN was offered to patients whose renal tumors were either solitary, solid masses or complicated renal cysts preoperatively classified as Bosniak type III or IV cysts, which were not located in the renal hilum or associated with central sinus invasion if patients had a normal contralateral kidney. After the operation, the pathological features were reviewed by an experienced pathologist according to the World Health Organization classification system. Angiomyolipomas (AMLs), oncocytomas, benign cystic nephromas, adenomas, hydatid cysts, and various other non-malignant lesions were classified as benign, whereas malignant lesions comprised clear cell, chromophobe, papillary, collecting duct, sarcomatoid variant, and multilocular cystic RCC. A total of 195 patients were identified who had undergone elective PN for a solitary renal mass with the intention of curing presumed RCC, and these patients were included in this retrospective study. During the study period, no patients had undergone surgery with the preoperative diagnosis of a benign lesion or urothelial carcinoma. No patients with a renal mass or metastases underwent biopsy or in vivo ablation. Consequently, we have not recommended active surveillance for small renal tumors presumed to be RCC. Furthermore, no patients had a known genetic predisposition to RCC or AMLs (such as von Hippel-Lindau disease, Birt-Hogg-Dube syndrome, or tuberous sclerosis). All operations were performed using the laparoscopic surgical technique with a transperitoneal or retroperitoneal approach based on the patient’s history of abdominal surgery, habitus, tumor location, and surgeon preference. Furthermore, for better preservation of renal function, the off-clamp technique was chosen in all cases. Signed informed consent was obtained from all the patients for the publication of this study and for processing their medical data.

## 3. Results

Among the 195 kidney lesions removed, 30 (15.4%) of them were classified as benign by the pathologist. Table 1 shows the demographics and pathological features of all renal masses. Considering the 30 patients with a benign renal mass, in one case, conversion to open surgery was required owing to uncontrollable bleeding. In two other cases, it was necessary to place a double-J pyeloureteral stent for the urinary fistula during the postoperative period. Consequently, the complication rate was 10% (3/30 cases) among patients diagnosed with a benign renal mass. The blood loss ranged from 150–1800 mL (mean: 523 mL), whereas the operation time ranged from 75–330 min (mean: 186 min). The pathological results (in decreasing order of incidence) were: (1) oncocytoma (*n* = 26 cases; 86.8%) and (2) angiomyolipoma (*n* = 2 cases; 6.6%) and renal cysts (*n* = 2 cases; 6.6%).

## 4. Discussion

Previously, urologists claimed that >90% of solid kidney lesions were RCC at surgery. However, according to daily results, after surgical treatment, up to 27% of suspected kidney lesions are identified as benign tumors on final histological examination; this incidence rate increases discernibly as the tumor size decreases [16,17,18,19]. McKiernan et al. [9], assessing a series of PNs performed over a period of more than 10 years, showed that a diagnosis of a benign tumor was made in 34 out of 281 patients (23%), with a median tumor size of 2.7 cm. In another study performed by Gill et al. [12], after the PN of 200 renal masses (both with laparoscopic and open techniques), 23% of these patients were identified as having benign tumors. Pahernik et al. [10], assessing > 500 PNs for kidney lesions with a mean diameter of 3 cm, recorded a benign percentage in 24% of the cases. The likelihood of the tumor being benign was found to be greater when the kidney mass was small and solitary; therefore, in the present study, we have retrospectively reviewed all PNs for a solitary renal mass performed at our department. Patients with a known genetic predisposition to RCC or those who underwent resection of more than one lesion were excluded. Our analysis revealed that 15.4% of the PNs performed for a suspected solitary RCC revealed the presence of a benign tumor (30/195 patients). In this scenario, we have to consider two important aspects. First is the fact that the clinical manifestations of these incidental masses were either absent or non-specific. Second, and more important than the first aspect, is the role of renal biopsy. In a review of 20 studies, including 2979 patients and 3113 biopsies, Patel et al. [20] reported that core biopsies were highly sensitive and specific when a diagnostic result was obtained. However, approximately 80% of patients did not undergo surgery following the benign biopsy result. After PN, 36.7% of patients with negative biopsy results showed malignant disease on surgical specimens. To overcome this problem, without forgetting the importance of tumor size in predicting whether a suspected renal mass is benign or not, we believe that other clinical factors (such as patient sex, patient age, and tumor location) can be evaluated as predictors of benign disease in patients with solitary solid renal masses, helping the urologist better select which patients may be suitable for renal mass biopsy. In this way, we believe that the rate of negative biopsy results, despite the presence of malignant renal tumors, could be decreased. With this in mind, we are carrying out a retrospective analysis of the ability of these clinical factors to predict the benignity or otherwise of a suspected renal lesion. Undoubtedly, imaging studies fulfill a fundamental role in evaluating small renal masses (SRMs). CT scans are currently the most commonly used imaging technique for the initial diagnosis and staging of suspected kidney lesions [21]. In adults, both malignant (such as RCC) and benign (such as AML and oncocytoma) kidney tumors may present as a solid mass. On performing a CT, a renal mass is generally considered non-enhancing if the change in attenuation is ≤10 Hounsfield units (HU) comparing the non-enhanced and contrast-enhanced scans or enhancing if the change is >20 HU. However, should a renal mass show a borderline enhancement (with a change of 10–20 HU), it is then suspected to be RCC [22,23]. In addition, on the one hand, due to their size, it may be difficult to assess the presence of enhancements in small renal lesions. On the other hand, some small RCCs, in particular papillary RCCs, show a low-level enhancement, and for this reason, these masses could be misidentified as hyperdense cysts [24]. On performing CT, macroscopic fat has an attenuation of <10 HU [25,26], and its presence in a non-calcified renal lesion is specific for a diagnosis of AML. In the majority of cases, this benign mass does not need to be treated, except when the volume is high (usually >4 cm, due to the increased risk of bleeding) or the patient complains of symptoms. Almost always on CT scans, the fat of AMLs is readily discernible, but if present in only small amounts, this may be obscured on a contrast-enhanced scan. Therefore, in these cases, performing an unenhanced scan with thin slice sections is useful [26]. AMLs without macroscopic or visible fat on imaging, denoted as “lipid-poor AMLs,” mimic RCCs; in addition, in very rare cases, macroscopic fat can be present in RCCs for a host of different causes: (i) engulfment of adjacent fat; (ii) osseous metaplasia [27]; or (iii) cholesterol necrosis [28]. These “fat-containing” RCCs represent a serious problem in terms of making a differential diagnosis with benign AML. In cases of equivocal enhancement in a renal lesion on performing CT, MRI should be considered a problem-solving technique [21]. In our series, two AMLs were surgically removed from the patients. Therefore, as highlighted in Table 1, the entire occurrence of AMLs of only 6.6% in our study is lower than the percentages previously reported, which provides a potential explanation for the lower rate of detection of non-malignant tumors in the current study. To date, when using common imaging techniques, there are no defining characteristics that can help us to discern oncocytomas from RCC. On a CT, in approximately one-third of cases, oncocytomas manifest themselves as well-capsulated solid lesions with a central scar; however, this feature is also observed in RCCs [21]. Several series have investigated whether enhancement behavior can be predictive of tumor histology. According to several of these studies, oncocytomas reveal a “segmental enhancement inversion” pattern during the corticomedullary and early excretory phases [29,30]. However, the same also applies to RCCs [31,32,33]. In the present study, 26 oncocytomas were resected. With common imaging techniques, the kidney masses that are most commonly identified are simple cysts that do not require any kind of treatment. According to the data in the previously published literature, at PN, the impact of complex renal cysts varies from 2–14% [34]; this range probably arises as a consequence of the different ways in which the authors of these studies defined a Bosniak type III cyst. The Bosniak classification stratifies cystic lesions into surgical and non-surgical types [35]. According to several series, ≤50% of Bosniak type III cysts are not benign [36,37]. Nevertheless, Marszalek et al. [3] showed a benignity rate of 92% among Bosniak type III cysts following nephron-sparing surgery. After surgical treatment, some of the Bosniak class III cysts are found to be benign on final pathological examination; however, by definition, no imaging techniques can generate results that allow us to distinguish these lesions from RCC. By contrast, Bosniak class IV cysts exhibit solid enhancing components and almost always have a malignant nature at surgery [35]. In the current study, two renal masses (6.6%) were identified as benign cysts at surgery. The relatively high occurrence of non-malign tumors after radical treatment indicates that proceeding directly to surgery should be avoided whenever possible, including with the use of mildly intrusive methods, such as laparoscopic nephron-sparing surgery. Some recently published surveys have reported low complications and a lack of tumor seeding following a biopsy of renal masses. Renal lesions, if small, usually increase and evolve tardily, especially over a short time period. Therefore, active surveillance has been chosen for patients who are at greater risk and for whom it would be better to avoid surgery. Especially for renal masses with a diameter of 2 cm or less, due to the higher incidence of benign pathology, active surveillance may obviate the need for more unnecessary surgical treatments, with its consequent morbidity. A meta-analysis of 10 published studies has evaluated the evolution of 234 small solid lesions that were not immediately resected [38]. In this meta-analysis, the authors did not identify any differences in the cancer dimension, neither at the initial nor the cancer evolution stages among oncocytomas and RCCs. Even when the impact of a deadly tumor was great, only three patients (1%) developed metastatic illness during the period of supervision. Incidentally, not every kidney cancer is associated with a pain-free path. In a meta-analysis of 6471 SRMs by Kunkle et al. [39], no differences were observed in the incidence of metastatic progression, regardless of whether the small renal mass was excised, ablated, or observed. Subsequently, according to Jewett et al. [40], delaying treatment until the small renal mass should progress had no adverse effects on the oncological outcome. Owing to the metastatic potential and several comorbidities, a safety cutoff size for active surveillance is still lacking [38]. Two different studies [38,40] reported a 12% incidence rate of local progression, and an approximate 1% rate of metastatic progression, during the first two years of follow-up. Patients who choose active surveillance must be informed of this low but non-negligible risk of progression. Percutaneous surgeries, including cryoablation and radiofrequency ablation, are also used as an alternative option to PN. Even though metastasis and not noticing RCC have occurred in a relatively small number of subjects going through cryoablation and radiofrequency ablation [41,42], the occurrence of regional relapse following cancer ablation has been shown to be higher compared with that following nephrectomy (both PN and radical nephrectomy), highlighting the relevance of careful employment of these latest methods [42]. In recent years, several authors have tried to find new methods that can facilitate the distinction between benign and malignant kidney tumors. Relatedly, blood oxygen level-dependent magnetic resonance imaging (BOLD-MRI) is to be considered; it is a kind of noninvasive MRI technology that reflects the tissue blood oxygen levels. In their study, Yelan Deng et al. [43] aimed to explore the value of radiomics based on BOLD-MRI in differentiating malignant from benign renal tumors. A total of 141 patients with renal tumors confirmed by pathology were retrospectively analyzed. In all patients, 118 with malignant tumors and 23 with benign tumors were confirmed. All the patients underwent renal T_1_-weighted imaging (T_1_WI), T_2_-weighted imaging (T_2_WI), and BOLD-MRI scan within the 2 weeks before surgery. The patients were randomly assigned into a training group and a test group. Two radiologists, who were blind to the pathological results, delineated the regions of interest (ROI) on the maximum axial slices of the tumors, and the radiomics signature score (Radscore) of each case was calculated using the Wilcoxon test to compare the difference in the Radscore between benign and malignant renal tumors in the training and test groups. The result was that the Radscores of malignant tumors in the training and test groups were higher than those of benign tumors, demonstrating that BOLD-MRI-based radiomics can be a reliable noninvasive approach for differentiating malignant renal tumors from benign tumors. Moreover, in differentiating malignant and benign renal tumors, Yuqin Ding et al. [44] quantitatively compared the diagnostic values of conventional diffusion-weighted imaging (DWI), intravoxel incoherent motion (IVIM), and diffusion kurtosis imaging (DKI). From the analysis of 180 patients affected by renal tumors, the result was that the IVIM parameters are the best, while DWI and DKI parameters have similar performance in differentiating malignant and benign renal masses. The use of DKI as another noninvasive biomarker for benign and malignant renal tumors’ differential diagnosis is suggested by other studies [45]. Other authors, like Hongtao Zhang et al. [46], assessed the usefulness of morphological characteristics of diffusion-weighted imaging (DWI) for differentiating malignant renal tumors from benign renal tumors and clear cell renal cell carcinoma (RCC) from non-clear cell RCC at 3.0 T. In his study, Zhang [46] included 249 patients with 251 histopathologically confirmed renal tumors that showed high signals on DWI. For each tumor, two radiologists independently evaluated apparent diffusion coefficient (ADC) values and morphological characteristics of DWI. The ADC values for malignant renal tumors were statistically significantly higher than those for benign renal tumors, and the ADC values for clear cell RCC were statistically significantly higher than those for non-clear cell RCC. The conclusion of this report is that the morphological characteristics of DWI are useful in differentiating renal tumors. In the field of imaging techniques, promising results also derived from the use of the aorta-lesion-attenuation difference (ALAD) determined on CT scan, which discriminated well between chromophobe RCC and oncocytoma. In this regard, Grajo et al. [47] tried to validate this evidence in a second cohort of patients undergoing nephrectomy for small solid renal masses by performing a retrospective review of preoperative CT scans and surgical pathology. ALAD was calculated by measuring the difference in Hounsfield units (HU) between the aorta and the lesion of interest on the same image slice on a preoperative CT scan. ALAD values were calculated during the excretory and nephrographic phases. The ability of the ALAD to discriminate between chromophobe RCC and oncocytoma was diminished in the validation cohort compared to the training cohort but remained significant. These findings support further investigation of the role of ALAD in the management of patients with indeterminate diagnoses of renal neoplasm. Yet, pointing out the clinical need for improved tools to aid in the pretreatment characterization of renal tumors to inform patient management, Eduard Roussel et al. [48] reviewed the evidence on noninvasive imaging-based tools for solid renal mass characterization by performing a systematic search for relevant studies on the MEDLINE database in the past 10 years. According to the authors, technologies of this particular note include multiparametric magnetic resonance imaging of the kidney, molecular imaging with targeted radiopharmaceutical agents, and the use of radiomics, as well as artificial intelligence to enhance the interpretation of imaging studies. Among these, 99mTc-sestamibi single photon emission computed tomography/computed tomography (CT) for the identification of benign renal oncocytomas and hybrid oncocytic chromophobe tumors, and positron emission tomography/CT imaging with radiolabeled Girentuximab for the identification of clear cell renal cell carcinoma, are likely to be closest to implementation in clinical practice. Also, with regard to the search for biomarkers, the scientific literature on renal cancer has made significant progress in recent years. A promising biomarker is ELABELA, whose expression in benign and malignant renal tissues and expression differences in different nuclear grades of clear cell carcinomas was investigated by G. Artas et al. [49]. In this study, the sample was composed of patients that underwent surgery due to renal masses. Control renal tissues, papillary RCC, clear cell RCC (CcRCC) (also considering the different Fuhrman Grade1), and chromophobe RCC were included in the study. ELA immunoreactivity was observed in proximal and distal tubules in the kidney but not in glomeruli in control tissues. When compared with control kidney tissue, a statistically significant increase was observed in ELA immunoreactivity in renal oncocytoma. In the chromophobe RCC, ELA immunoreactivity was significantly lower than in control kidney tissue, whereas papillary RCC did not show ELA immunoreactivity. However, compared with control kidney tissue, ELA immunoreactivity was not observed in Fuhrman Grade 1 and Grade 2 CcRCC. Also, there was a significant decrease at Fuhrman Grade 3 and Grade 4 CcRCC compared with control kidney tissues. In the statistical analysis of ELA immunoreactivity among the Fuhrman nuclear grades of CcRCCs, the ELA immunoreactivity was higher at Grade 4 CcRCC than at Grade 1, Grade 2, and Grade 3. According to the results of this study, ELA can be a useful molecule to differentiate benign and malign renal tumors. However, further broad and comprehensive studies are needed to investigate cellular and molecular mechanisms of ELAs on malign transformation. Also, the role of the forkhead box 1 (FOXI1) transcription factor is investigated in the differential diagnosis of renal cancer, especially in the distinction between Renal Oncocytoma (RO) and chromophobe renal cell carcinoma (chRCC). These tumors are suggested to develop from alpha- and beta-intercalated (IC) cells of the collecting duct expressing solute carrier family 4 member 1 (SLC4A1) and SLC26A4 under the control of FOXI1. In this regard, in their study, Molnar et al. [50] aimed to clarify the possible cellular origin of RO and chRCC by immunohistochemistry for aquaporin 2 (AQP2), FOXI1, SLC4A1, and SLC16A4. Nuclear FOXI1 staining occurred in 96% of 83 ROs, in 3% of 90 chRCCs, and in none of the other tumor types. So, although the origin of RO remains unclear, their findings suggest that FOXI1 immunohistochemistry is useful in the differential diagnosis of RO from chRCC with overlapping histology. Despite relevant discoveries both in the field of biomarkers and imaging techniques in the differential diagnosis of renal cell carcinoma, PN remains not only a fundamental surgical method in the treatment of small renal cancer but also an important diagnostic tool. In our series, 6.6% of benign renal tumors were AML, 86.6% were oncocytoma, and 6.6% were cystic masses. The exact proportion of histological types varies across several studies. For example, in one series [51], >80% of the benign kidney tumors were oncocytomas. In other series [18,52,53], however, oncocytoma was diagnosed less commonly. Clearly, selection bias has an impact on the proportion of histological types. It should be noted that the present study had a number of limitations. First, sorting misconceptions may have existed since the study was a retrospective one, and second, the patients included were all subjects of a single center. Despite this, the procedures for elective PN at our department were the same as those of other centers, and the study sample is relevant, considering that it is a mono-centric report.

In conclusion, the present study has shown the incidence rate of benign tumors in patients who have been subjected to laparoscopic partial nephrectomy due to a suspected solitary renal mass. Based on these results, the patient should be counseled not only about the intra- and post-operative risks of nephron-sparing surgery but also about its dual therapeutic and diagnostic role. Therefore, patients should be informed about the considerably high probability of a benign histological result. Furthermore, considering the crucial problem of the socioeconomic burden of PN and its associated complications in patients with benign kidney tumors (where a complication rate of 10% was noted in our study among patients diagnosed with benign renal mass), it is clear that urologists need to focus on trying to reduce non-malignant final pathological diagnoses.

## Figures and Tables

**Table 1 diseases-11-00027-t001:** Demographics and pathological features of all renal masses.

Gender total population (195 patients), *n* (%)	*Female*	*Male*		
	82 (42.1%)	113 (57.9%)		
Total population Age, mean (SD), years	60.9 years (from 29.9 to 79 years)			
Histological type: malignant tumors, *n* (%)	*RCC*	*Papillary*	*Cromophobe*	*Others*
	126 (76.4%)	21 (12.7%)	18 (10.9)	-
Histological type: benign tumors, *n* (%)	*Oncocytoma*	*Angiomyolipoma*	*Renal Cyst*	*Others*
	26 (86.8%)	2 (6.6%)	2 (6.6%)	-
Malignant Tumor size, mean (SD), cm	5.7			
Benign Tumor size, mean (SD), cm	4.4			
Sex distribution of benign tumors	*Female*	*Male*		
	11	19		
Sex distribution of malignant tumors	*Female*	*Male*		
	52	113		
Side distribution of benign tumors	Right Kidney	Left Kidney		
	15	15		
Renal site distribution of benign tumors	*Upper pole*	*Middle site*	*Lower Pole*	
	15	9	6	

## Data Availability

The datasets used and/or analyzed during the current study are available from the corresponding author on reasonable request.

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
