# Peer review of "“To Be or Not to Be Benign” at Partial Nephrectomy for Presumed RCC Renal Masses: Single-Center Experience with 195 Consecutive Patients"

_diseases, 2023, doi:10.3390/diseases11010027_

Round 1
Reviewer 1 Report
Although it is a well-presented article, the authors failed to provide new points to this subject. In fact, the percentage of benign renal tumors among the surgical samples is well documented. The authors correctly discussed every point that has been already known. The authors should provide new points to this subject. Whar are recent advances concerning this subject? There are almost no reference article of recent years.
Author Response
We thank the reviewer for appreciating our manuscript, judging it well written and well documented. We thank the reviewer for the valuable suggestion to provide new points to this subject. In this regard we have expanded the discussion with reference to recent discoveries in the field of renal tumor diagnostics, with references to articles published less than five years ago.
Reviewer 2 Report
The work is retrospective and should be considered as such. The series is numerically good in relation to that of other works reported in the literature and the results of your study presented are in line with them.
It might be interesting also to include in the discussion your more specific point of view on whether or not it is necessary to biopsy small renal masses suggestive of benignity.
This is a challenge according to the percentage of biopsies that are truly positive for benign masses (after PN about 37% of patients with negative biopsy were malignant -21).
Author Response
We thank the reviewer for appreciating our article and for the important suggestion to include in the discussion our more specific point of view on whether or not it is necessary to biopsy small renal masses suggestive of benignity. We did so, highlighting this integration.
Reviewer 3 Report
1. In the introduction, it is better not to use tables (Table 1) and provide relevant information in the form of text.
2. In general, the clinical guidelines indicate that it is recommended to perform kidney resection, including laparoscopic, in patients with clinically localized forms of kidney cancer (stages T1-T2N0M0) in cases where the size and localization of the tumor allow performing organ-preserving operations. In any case, the patient cannot make a decision on his own. I do not quite understand the purpose of the study. The authors could compare the CT scan data and predict which tumors might be benign or something else. The article seems unfinished to me.
3. Kidney resection was compared with nephrectomy in the only a randomized phase III trial conducted by the European Organization for Research and Treatment of Cancer Genito-Urinary Group (EORTC-GU), designed as noninferiority ("not worse") and prematurely closed due to poor patient recruitment (541 of 1300 planned recruits) . The study included patients with renal parenchymal tumors <4 cm convenient for organ-preserving intervention. The analysis of the results was carried out according to the randomization group and showed a trend towards a decrease in OS in the group with kidney resection compared with nephrectomy at a median follow-up of 9.3 years (odds ratio (OR) 1.5; 95% confidence interval (CI) 1.03–2.16; p = 0.03).
A Canadian meta-analysis (2016) of individual data from patients with T1N0M0 renal parenchymal cancer included in the Canadian Kidney Cancer Information System who underwent kidney resection (n = 1615) or nephrectomy (n = 2358) did not reveal significant differences in time to progression in groups (OR 1.17, 95% CI 0.8–1.72, p = 0.42).
Van Poppel H., da Pozzo L., Albrecht W. A prospective, randomised EORTC intergroup phase 3 study comparing the oncologic outcome of elective nephron-sparing surgery and radical nephrectomy for low-stage renal cell carcinoma. Eur Urol 2011;59(4):543−52. DOI: 10.1016/j.eururo.2010.12.013.
Kunath F., Schmidt S., Krabbe L.M. Partial nephrectomy versus radical nephrectomy for clinical localized renal masses. Cochrane Database Syst Rev 2017;5:CD012045. DOI:10.1002/14651858.CD012045.pub2.
Forbes C.M., Rendon R.A., Finelli A. Disease progression and kidney function after partial vs. radical nephrectomy for T1 renal cancer. Urol Oncol 2016;34(11):486.e17−23. DOI:10.1016/j.urolonc.2016.05.034.
Mir M.C., Derweesh I., Porpiglia F. Partial nephrectomy versus radical nephrectomy for clinical t1b and t2 renal tumors: a systematic review and meta-analysis of comparative studies. Eur Urol 2017;71(4):606−17. DOI: 10.1016/j.eururo.2016.08.060.
Yu-Li J., Cheng-Xia P., Heng-Zi W., Lu-Jie Q. Comparison of the long-term follow-up and perioperative outcomes of partial nephrectomy and radical nephrectomy for 4 cm to 7 cm renal cell carcinoma : a systematic review and meta-analysis. BMC Urol 2019;19(48). DOI: 10.1186/s12894-019-0480-6.
I recommend that the authors familiarize themselves with these works and revise the manuscript.
Author Response
1) We thank the reviewer for the important suggestion to include the information contained in Table 1 in text form. We did so, eliminating the table and reporting the information in the text.
2) The purpose of our study is to investigate the incidence of benign tumors at the stage of receiving PN on preoperative imaging evaluations in a single-center series of patients with a solitary renal lesion considered to be RCC. This is a scientific research already addressed by other authors but we believe we can strengthen the statistical evidence with a case study that is decidedly relevant since it is a single center study. Regarding the comparison with CT scan data, we have addressed this topic in the discussion, providing a broad overview on the ability or not to pre-operatively diagnose a suspected renal lesion as malignant or not.
3)
We thank the reviewer for the suggestion to review the articles he reported. However, since the cited articles aim to compare the oncological outcomes of patients treated with partial or radical nephrectomy, we believe that they are far from the scope of our research. Our article aims to evaluate the rate of benignity for patients who are candidates for partial nephrectomy for renal lesions suspected of malignancy, offering a broad and current examination of the problem of subjecting the patient to surgery with its intrinsic complications faced with the risk of removing a lesion which later proves to be benign.
Round 2
Reviewer 1 Report
The authors have considerably expanded the discussion on recent discoveries in the field of renal tumor diagnosis in their revision. The article may be accepted.
Reviewer 3 Report
The authors explained their position to the reviewer, I leave them the right to their point of view. In its present form, the manuscript can be recommended for publication.